https://doi.org/10.1038/s42003-022-04130-0　　**OPEN**
# Paraquat is an agonist of STIM1 and increases intracellular calcium levels

Wenyu Yang[1,5], Rui Tian[1,5], Yong Zhu[1,5], Peijie Huang[1,5], Xinrun Ma[2], Xiaoxiao Meng[1], Wentao Dai[3], Yiming Tao[4], Daonan Chen[1], Jiaxiang Zhang[1], Jian Lu[1], Hui Xie[1], Xiangdong Jian[4,6✉], Zhengfeng Yang[1,2,6✉] & Ruilan Wang[1,6✉]

Paraquat (PQ) is an efficient herbicide but leads to high mortality with no antidote in mammals. PQ produces reactive oxygen species (ROS), leading to epithelial-mesenchymal transition (EMT) for pulmonary fibrosis in type II alveolar (AT II) cells. Intriguingly, strategies reducing ROS exhibit limited therapeutic effects, indicating other targets existing for PQ toxicity. Herein we report that PQ is also an agonist for STIM1 that increases intracellular calcium levels. Particularly, PQ promotes STIM1 puncta formation and association with TRPC1 or ORAI for extracellular calcium entry and thus intracellular calcium influx. Further studies reveal the importance of P584&Y586 residues in STIM1 for PQ association that facilitates STIM1 binding to TRPC1. Consequently, the STIM1-TRPC1 route facilitates PQ-induced EMT for pulmonary fibrosis as well as cell death. Our results demonstrate that PQ is an agonist of STIM1 that induces extracellular calcium entry, increases intracellular calcium levels, and thus promotes EMT in AT II cells.

[1] Department of Critical Care Medicine, Shanghai General Hospital, Shanghai Jiaotong University, School of Medicine, Shanghai 201620, China. [2] Precision Research Center for Refractory Diseases, Shanghai General Hospital, Shanghai Jiaotong University, School of Medicine, Shanghai 201620, China. [3] Shanghai Center for Bioinformation Technology, Shanghai Academy of Science and Technology, Shanghai 201203, China. [4] Department of Poisoning and Occupational Diseases, Emergency, Qilu Hospital, Cheeloo College of Medicine, Shandong University, Jinan, Shandong 250012, China. [5] These authors contributed equally: Wenyu Yang, Rui Tian, Yong Zhu, Peijie Huang. [6] These authors jointly supervised this work: Xiangdong Jian, Zhengfeng Yang, Ruilan Wang. ✉email: jianxiangdongvip@vip.163.com; Zhengfeng.yang@shgh.cn; wangyusun@hotmail.com

PQ is a highly efficient herbicide commonly utilized worldwide but with extreme toxicity to mammals. More than 30 countries have forbidden the production or utilization of PQ since its introduction. However, PQ has still been utilized in around 100 countries and regions for agricultural production, including the United States, Japan, Australia, New Zealand, etc.[1,2]. PQ can be absorbed by multiple organs, including the digestive tract, respiratory tract, and skin, and leads to organ failure, especially pulmonary fibrosis. Unfortunately, no antidote so far has been utilized in clinic for PQ poisoning[3,4]. PQ has been well characterized to play a catalytic role in the redox cycling process that leads to the production of ROS[5]. Multiple downstream cascades of ROS signaling have been reported to mediate PQ toxicity, including DNA damage, abnormal accumulation of mitochondrial ROS, mitochondrial dysfunction, inflammation, and protease imbalance, followed with epithelial-mesenchymal transition (EMT) and severe cell death in alveolar type II (AT II) cells, which together promote pulmonary fibrosis and lung injury[6–8]. Intriguingly, clinical treatments based on these mechanisms are not available to efficiently prevent or treat PQ-induced pulmonary fibrosis and following high mortality[2,9], indicating that molecular targets other than the redox cycling process exist to mediate PQ toxicity, which requires further efforts to elucidate. We and others have revealed that PQ-induced pulmonary fibrosis is largely based on the accumulation of ROS and the induction of EMT in AT II cells[6–8]. Store-operated calcium channel (SOC)-mediated extracellular calcium ($Ca^{2+}$) entry, including store-operated calcium entry (SOCE), promotes intracellular $Ca^{2+}$ burden, which has been well established as an important upstream signal for ROS and EMT[10,11]. Fan et al., further revealed that PQ increases the intracellular $Ca^{2+}$ concentration in 16HBE cells, the human bronchial epithelial cells[12], which together raise the possibility that PQ could activate SOC for pulmonary fibrosis. Therefore, we assumed that PQ initiates both ROS production and intracellular $Ca^{2+}$ overload in AT II cells, which then aggravates pulmonary fibrosis and PQ poisoning, making a simple blockage of ROS signal during PQ poisoning exhibit limited efficacy.

Multiple factors modulate intracellular $Ca^{2+}$ overload, yet extracellular $Ca^{2+}$ is the sole source for intracellular $Ca^{2+}$, from which SOCE is well recognized as the major route for extracellular $Ca^{2+}$ influx in non-excitable cells. Modulation of intracellular $Ca^{2+}$ levels by SOCE facilitates multiple patterns of $Ca^{2+}$ signal, making $Ca^{2+}$ be a versatile second messenger for numerous cellular functions[13,14]. Stromal Interaction Molecule 1 (STIM1) and ORAI calcium release-activated calcium modulator 1 (ORAI1) are two major components for SOCE activation. Mutations in these two components have been identified to drive severe disorders in patients due to abnormal intracellular $Ca^{2+}$ signal[15]. Activation of SOCE is a fine-tune modulated multistep process, including endoplasmic reticulum (ER) $Ca^{2+}$ content depletion due to activation of inositol 1,4,5-trisphosphate receptors or inhibition of sarcoendoplasmic reticulum calcium ATPase pumps[16], STIM1 oligomerization and redistribution, STIM1 association with ORAI1, and STIM1 deoligomerization after ER $Ca^{2+}$ refilling[17–20]. In certain scenarios, STIM1 also associates with TRP family proteins and forms store-operated calcium channel (SOC), which the poly-lysine (K) region of STIM1 is critical for gating TRPs, including Transient Receptor Potential Cation Channel Subfamily C Member 1 (TRPC1)[21–23]. In pathological conditions, STIM1, SOCE, or SOC are overactivated, leading to intracellular $Ca^{2+}$ overload and multiple disorders. Though studies have shown a positive correlation between ROS production and intracellular $Ca^{2+}$ levels in the condition with PQ stimulation[12,24], whether PQ targets SOCE/SOC and therefore pulmonary fibrosis requires further efforts to clarify, which might provide potential efficient strategies for treating PQ poisoning.

We recently showed that the $Ca^{2+}$ signal could be the mechanistic difference in toxicity between PQ and diquat, another efficient herbicide sharing chemical backbone similar to that of PQ[25], raising the potency of $Ca^{2+}$ signal in mediating PQ toxicity. Herein, we further aim to study whether and how PQ modulates intracellular $Ca^{2+}$ signal and the relevant molecular targets of PQ for pulmonary fibrosis. Our results demonstrate that PQ is an agonist of STIM1 that promotes STIM1-mediated calcium entry and promotes EMT and cell death in AT II cells, which would aid the precise understanding of PQ toxicity and the development of potential clinical strategies for PQ poisoning in the future.

## Results

### PQ promotes intracellular $Ca^{2+}$ overload and NFAT activation.
To address whether PQ modulates intracellular $Ca^{2+}$ levels, we first measured intracellular $Ca^{2+}$ levels by flow cytometry. Our results showed that intracellular $Ca^{2+}$ levels are significantly increased in A549 cells exposed to PQ in a dose dependent manner for 24 h (Fig. 1a). Similar results were observed in MLE 12 cells, another AT II cell line (Fig. 1b). Conversely, the concentration of intracellular $Ca^{2+}$ in WI-38 cells, the lung fibroblast cells, after PQ treatment exhibits no significant difference compared to untreated groups (Fig. 1c). Single-cell $Ca^{2+}$ imaging further confirmed that PQ stimulation increases intracellular calcium levels in A549 cells (Fig. 1d). These data show that PQ specifically induces intracellular $Ca^{2+}$ overload in AT II cells. Previous studies have disclosed that repetitive or prolonged increase in intracellular $Ca^{2+}$ is required for NFAT activation[26]. We therefore utilized the NFAT-luciferase reporter system to evaluate whether PQ modulates the downstream activity of $Ca^{2+}$ signaling. As expected, PQ strongly raises the activity of NFAT (Fig. 1e) as well as the expression of NFATc1 and NFATc2 (Fig. 1f, g) after PQ treatment for 24 h. NFAT activation was further confirmed by its nuclear translocation, in which NFATc1 is thoroughly translocated from the cytosol into the nucleus after PQ treatment (Fig. 1h). Collectively, our data suggest that PQ poisoning raises intracellular $Ca^{2+}$ levels in AT II cells.

### PQ activates STIM1-mediated calcium entry to increase intracellular $Ca^{2+}$ levels.
Multiple $Ca^{2+}$ channels have been reported to modulate extracellular $Ca^{2+}$ entry, of which SOC and SOCE are the primary $Ca^{2+}$ entry pathways in non-excitable cells[27]. STIM1, ORAI1 and multiple TRPs are all expressed in A549 cells. Intriguingly, PQ treatment dramatically reduces the expression of TRPs but not STIM1 or ORAI1. Also, though largely reduced, TRPC1 is still highly expressed compared to other TRPs after PQ treatment (Fig. 2a). Therefore, we turn our sight on STIM1, ORAI1 and TRPC1 in the modulation of PQ-induced extracellular $Ca^{2+}$ entry and subsequent increase in intracellular $Ca^{2+}$ levels. By utilizing SKF96365 (abbreviated SKF), a SOCE inhibitor and a non-selective TRP Channel blocker, we found that PQ-increased intracellular $Ca^{2+}$ levels and NFAT-luciferase expression are both largely reduced by SKF co-treatment (Fig. 2b, c, Supplementary Fig. 1a). These results together indicate that PQ activates SOC or SOCE for the increase of intracellular $Ca^{2+}$ levels.

To clarify whether PQ does activate SOCE or SOC, we firstly measured STIM1 puncta formation under PQ treatment. As expected, PQ stimulation strongly induces STIM1 puncta formation in a dose dependent manner (Fig. 2d). Quantification of the area and perimeter of puncta further confirmed these results (Fig. 2e). Activation of STIM1 could be attributed to a reduction or depletion of ER $Ca^{2+}$ content. We then analyzed if PQ affects ER $Ca^{2+}$ content and therefore modulates STIM1 activation. Single-cell $Ca^{2+}$ imaging showed that transient PQ stimulation does not significantly affect $Ca^{2+}$ flux (Fig. 2f, left). Also, by utilization of thapsigargin (TG) stimulation to mimic ER $Ca^{2+}$ store depletion, we found that PQ pretreatment does not largely modulate ER $Ca^{2+}$

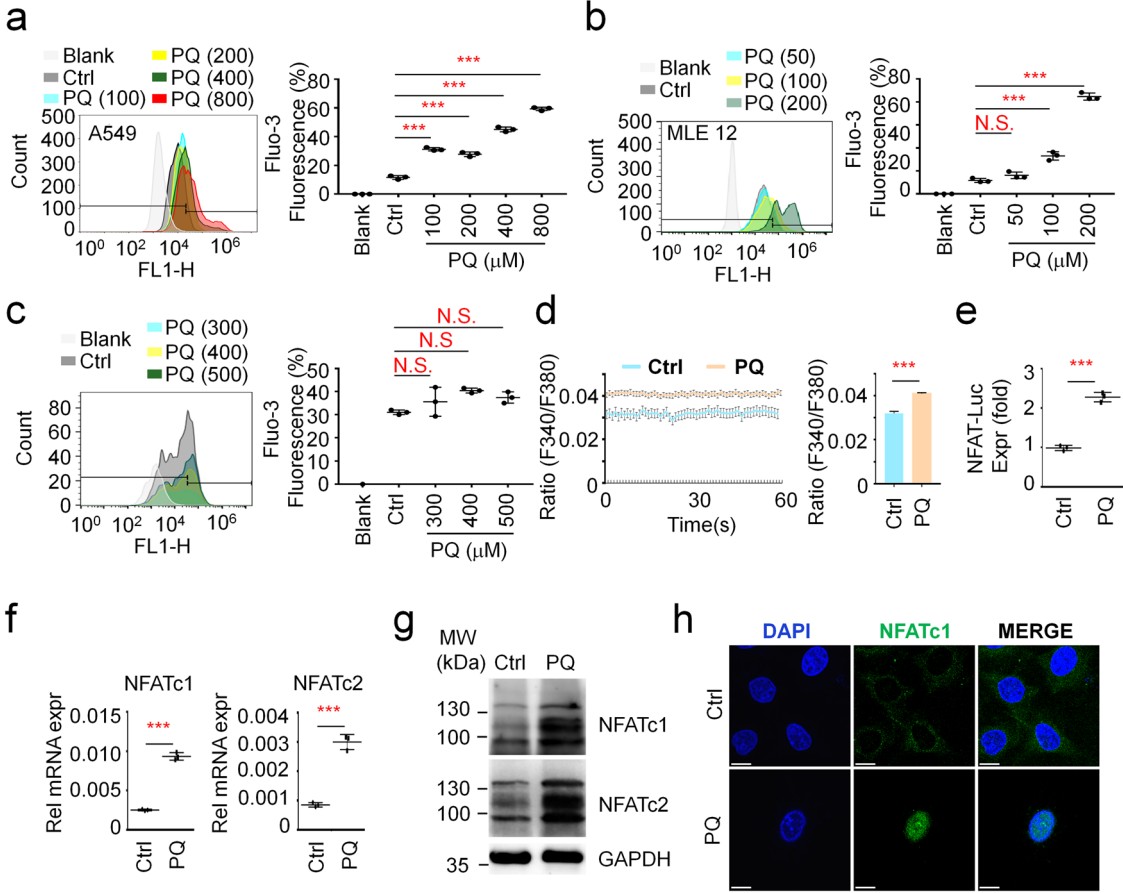

**Fig. 1 PQ promotes intracellular calcium overload and NFAT activation. a–b** FACS analysis of intracellular calcium levels by Fluo-3 staining in A549 cells (**a**) and MLE 12 cells (**b**). Cells were treated with or without (w/wo) PQ in a dose dependent manner as indicated for 24 hours. Mean ± s.d., ***$P < 0.001$; unpaired two-tailed Student's *t*-test. $n = 3$. **c** FACS analysis of intracellular calcium levels by Fluo-3 staining in WI-38 cells treated with 0, 300, 400, or 500 μM PQ for 24 hours. Mean ± s.d., $n = 3$. **d** Calcium imaging of A549 cells treated w/wo 800 μM PQ for 24 hours. Mean ± s.d., ***$P < 0.001$; unpaired two-tailed Student's *t*-test. $n = 58$. **e** NFAT luciferase expression in A549 cells treated w/wo 800 μM PQ for 24 hours. Mean ± s.d., ***$P < 0.001$; unpaired two-tailed Student's *t*-test. $n = 3$. **f** RT–PCR analysis to measure the expression of *NFATc1* and *NFATc2* in A549 cells treated w/wo 800 μM PQ for 24 hours. Mean ± s.d., ***$P < 0.001$; unpaired two-tailed Student's *t*-test. $n = 3$. **g** Western Blot to detect the expression of NFATc1 and NFATc2 in A549 cells treated w/wo 800 μM PQ for 24 hours. Images are representative of more than three experiments. **h** Immunofluorescence analysis of NFATc1 nuclear translocation in STIM1 stably expressing A549 cells treated w/wo 800 μM PQ for 24 hours. Scale bars, 10 μm.

contents or affect ER Ca²⁺ reuptake (Fig. 2f, right). We then further measured TG-raised extracellular calcium influx with or without PQ pre-stimulation, or PQ-induced calcium influx with or without TG pre-stimulation. Interestingly, we observed that PQ pretreatment enhances TG-induced SOCE (Supplementary Fig. 1b) while TG pretreatment does not affect PQ-induced extracellular calcium influx (Supplementary Fig. 1c). We considered that TG pretreatment leads to a longer window for TG to take effect, which depletes ER calcium stores and fully activates STIM1, making a later addition of PQ could not activate STIM1 anymore. Nevertheless, these results suggest that PQ activates STIM1 in a manner most likely independent of ER Ca²⁺ content.

STIM1 activation has been well established to promote SOCE or SOC. Indeed, PQ pretreatment remarkably facilitates the intracellular Ca²⁺ increase after exogenous Ca²⁺ was administered in a dose dependent manner (Fig. 2g). The route for PQ-induced extracellular Ca²⁺ entry could be due to either the STIM1-ORAI1 association or the STIM1-TRPC1 association. Co-immunoprecipitation analysis revealed that PQ largely promotes STIM1 association with ORAI1 or TRPC1 (Fig. 2h). Also, ectopic expression of STIM1, ORAI1 or TRPC1 all enhance PQ-raised intracellular calcium levels (Fig. 2i). To further confirm that PQ-raised extracellular calcium entry is modulated by STIM1, ORAI1 and TRPC1, we utilized another three

approaches. Firstly, it has been reported that the activation of STIM1-TRPC1 route also promotes Sr²⁺ influx[28], which can also be measured by Fura-2 staining. We then examined whether PQ stimulation facilitates Sr²⁺ influx. As shown in Supplementary Fig. 1d, PQ does promote Sr²⁺ influx (Supplementary Fig. 1d), indicating that PQ at least activates the STIM1-TRPC1 route. Secondly, we monitored PQ-induced extracellular calcium influx in cells with or without STIM1, ORAI1, or TRPC1 deficiency. As expected, compared to empty vector (EV) expressing cells, PQ-stimulated calcium influx is largely diminished in STIM1, ORAI1 or TRPC1 deficient cells (Fig. 2j) Finally, we examined whether the addition of SKF would suppress PQ-raised calcium influx. Consistent with other results, PQ-raised extracellular calcium influx is almost blocked by addition of 50 μM SKF (Fig. 2k). Taken together, these results suggest that PQ activates intracellular Ca²⁺ overload mainly via activation of STIM1 but independent of modulating ER Ca²⁺ content. In another word, PQ promotes STIM1-mediated calcium entry.

**PQ targets STIM1 and promotes the association between STIM1 and TRPC1.** Next, we wonder the precise mechanism of PQ in the modulation of extracellular calcium influx. As mentioned above, PQ might directly activate STIM1 and promote

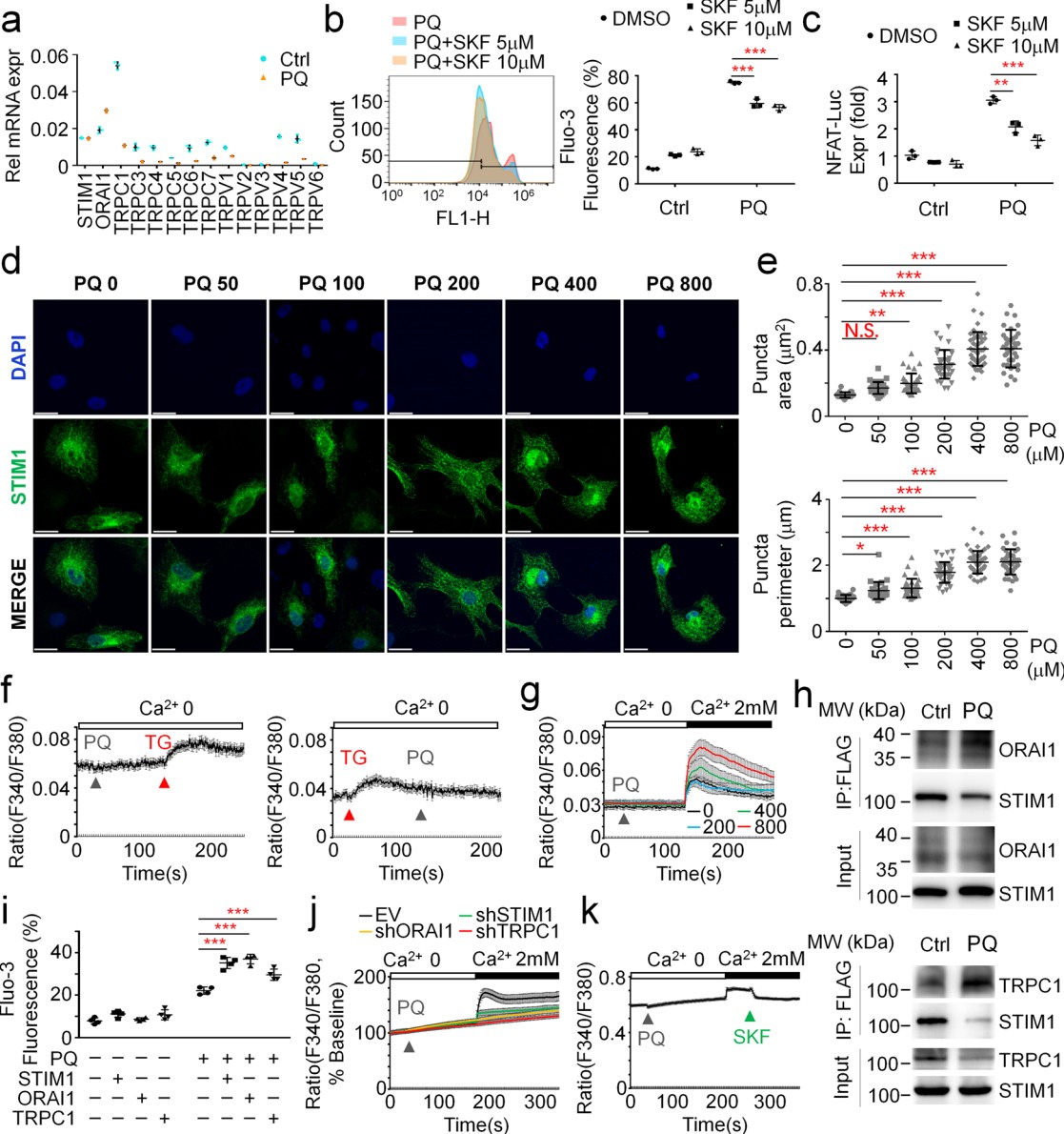

**Fig. 2 PQ induces extracellular calcium influx for intracellular calcium overload. a** RT–PCR analysis to detect the expression of *STIM1, ORAI1 and TRPs* in A549 cells treated w/wo 800 µM PQ for 24 hours. **b** FACS analysis of intracellular calcium levels by Fluo-3 staining in A549 cells. Cells were treated w/wo 800 µM PQ, together with 5 µM or 10 µM SKF-96365 (SKF) as indicated, for 24 hours. Mean ± s.d., ***$P < 0.001$; One-Way ANOVA. $n = 3$. **c** NFAT luciferase expression in A549 cells treated w/wo 800 µM PQ, together with 5 µM or 10 µM SKF-96365 (SKF) as indicated, for 24 hours. Mean ± s.d., **$P < 0.01$, ***$P < 0.001$; One-Way ANOVA. $n = 3$. **d–e** Representative images (**d**) and quantitative analysis (**e**) of STIM1 puncta observed by immunofluorescence in STIM1 stably expressing A549 cells treated with 0, 50, 100, 200, 400, or 800 µM PQ for 24 hours. Scale bars, 20 µm. **f** Calcium imaging of A549 cells treated with 800 µM PQ, addition with 1 µM thapsigargin; or 1 µM thapsigargin addition with 800 µM PQ. TG, thapsigargin. Mean ± sem. **g** Calcium imaging of A549 cells treated with PQ in a dose dependent manner as indicated, with the addition of 2 mM CaCl₂. Mean ± sem. **h** Co–immunoprecipitation (IP) analysis of STIM1 association with ORAI1 (left) or TRPC1 (right) in STIM1-FLAG stably expressing A549 cells treated w/wo 800 µM PQ for 24 hours. Images are representative of more than three experiments. **i** FACS analysis of intracellular calcium levels by Fluo-3 staining in A549 cells with ectopic expression of STIM1, ORAI1, or TRPC1, respectively. Cells were treated w/wo 800 µM PQ for 18 hours. Mean ± s.d., ***$P < 0.001$; Two-Way ANOVA. $n = 3$. **j** Calcium imaging of A549 cells infected with EV, shRNA-STIM1, shRNA-ORAI1, or shRNA-TRPC1, respectively. Cells were stimulated with 800 µM PQ, followed by 2 mM CaCl₂. Mean ± sem. **k** Calcium imaging of A549 cells stimulated with 800 µM PQ, followed by 2 mM CaCl₂, and 50 µM SKF96365. Mean ± sem.

STIM1 association with ORAI1 or TRPC1 for extracellular calcium entry. Interestingly, the association between STIM1 and TRPC1 but not the association between STIM1 and ORAI1 is largely enhanced by PQ stimulation in a short period (Fig. 3a,b). The C-terminal region of STIM1 has been reported to be important for the association with ORAI1 or TRPC1. Specifically, the SOAR region of STIM1 is mainly responsible for ORAI1

association[29–31], whereas the poly-K region of STIM1 is required for TRPC1 gating and their electrostatic associations[21,23]. We next introduced the Flag-tagged STIM1ΔK (deletion of poly-K region) mutant. PQ failed to promote the interaction between STIM1ΔK and TRPC1 or ORAI1 (Fig. 3c), suggesting the importance of the poly-K region of STIM1 in modulation of the PQ-activated STIM1-TRPC1 axis. As the poly-K region is

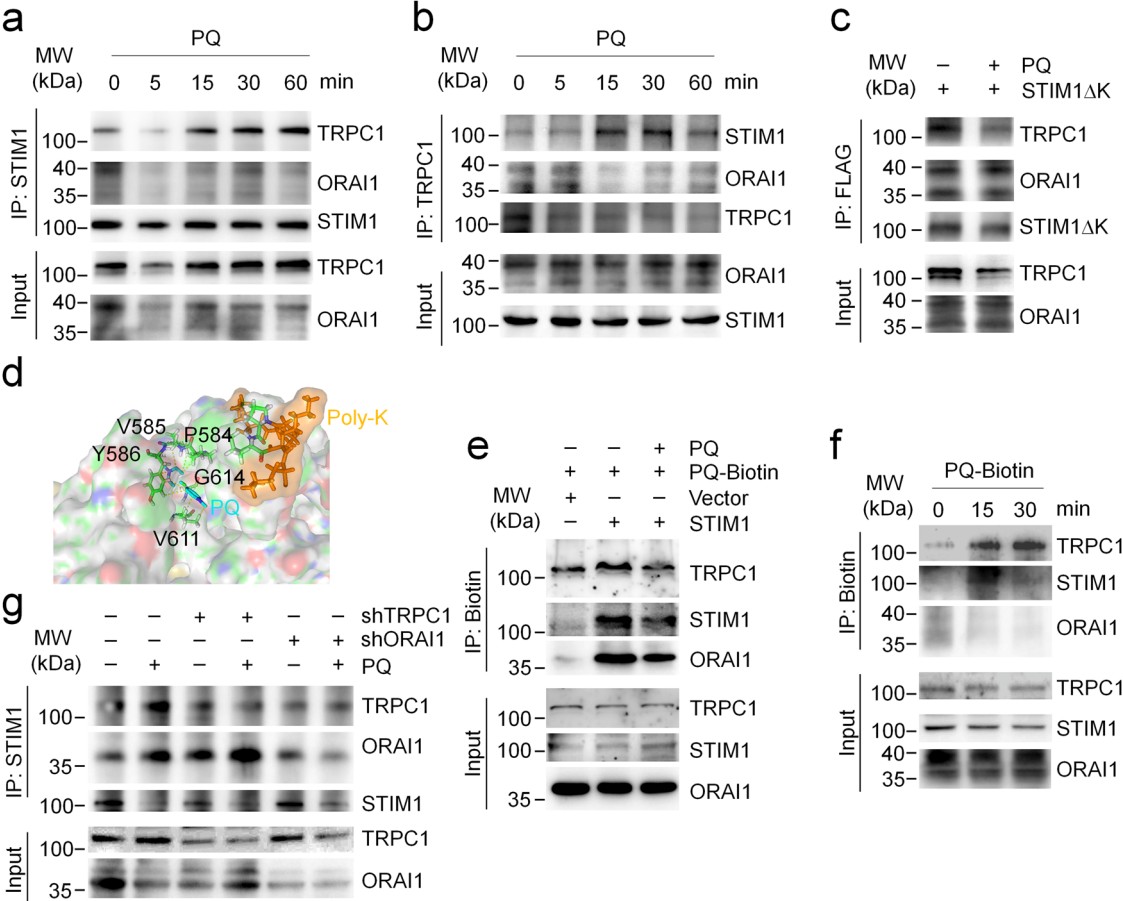

**Fig. 3 PQ binds to STIM1 and activates the STIM1-TRPC1 axis. a–b** Co-IP analysis of STIM1 association with ORAI1 or TRPC1 (**a**) or TRPC1 association with STIM1 or ORAI1 (**b**) in A549 cells treated with 800 µM PQ for 0, 5, 15, 30, and 60 min as indicated. Images are representative of more than three experiments. **c** Co-IP analysis of STIM1ΔK association with ORAI1 or TRPC1 in STIM1ΔK-FLAG stably expressing A549 cells treated w/wo 800 µM PQ for 24 hours. Images are representative of three experiments. **d** Molecular modeling of PQ and STIM1. The side view of the PQ-STIM1 complex in a surface representation showed potential residues in the C-terminal region of STIM1 forming a pocket for PQ that facilitates the exposure of the poly-K region. PQ is colored in cyan; potential residues of STIM1 for association with PQ is colored in green and the poly-K region of STIM1 is colored in yellow. **e–f** Pull-down analysis of PQ-Biotin and STIM1/ORAI1/TRPC1 association in A549 cells transfected to express empty vector or STIM1-GFP. Cells were treated with 800 µM PQ-Biotin for 24 hours (**e**) or 0, 15, 30 min as indicated (**f**). Immunoblot was probed using the indicated antibodies. **g** Co-IP analysis of STIM1 association with ORAI1 or TRPC1 in A549 cells infected with shRNA-ORAI1 or shRNA-TRPC1, respectively, and treated with 800 µM PQ for 24 hours.

required for STIM1 anchoring to the plasma membrane for association with ORAI1, the reduction of STIM1ΔK association with ORAI1 in response to PQ stimulation further indicates that the enhanced association between STIM1 and ORAI1 after long term exposure to PQ stimulation could be a by-product of STIM1 activation. The molecule modeling and docking analysis further indicates the potential amino acids for PQ association with STIM1 (Fig. 3d). This model raises us to assume that STIM1 is the molecular target for PQ. To address this assumption, we chemically modified the PQ molecule by linking with a biotin tag (Supplementary Fig. 1e). PQ-Biotin successfully associates with STIM1, ORAI1 or even TRPC1. Importantly, overexpression of STIM1 further enhances the association, whereas addition of the same amount of PQ as PQ-Biotin largely diminishes the association between PQ-Biotin and STIM1, TRPC1, or ORAI1 (Fig. 3e). A short time window analysis further revealed that PQ-Biotin associates with TRPC1 and STIM1 but not ORAI1 (Fig. 3f). These results again emphasize the importance of the STIM1-TRPC1 axis in mediating PQ toxicity. STIM1, ORAI1 and TRPC1 have been shown to work together to facilitate extracellular $Ca^{2+}$ entry, which the association between TRPC1 and STIM1 requires the existence of ORAI1[32]. We then examined the

association between STIM1 and ORAI1 or TRPC1 in the condition of TRPC1 deficiency or ORAI1 deficiency. TRPC1 deficiency does not affect PQ-induced STIM1 association with ORAI1 while ORAI1 deficiency dampens PQ-raised STIM1 association with TRPC1 (Fig. 3g), suggesting that PQ-induced STIM1 activation is independent of TRPC1 whereas ORAI1 is essential for PQ-stimulated STIM1 association with TRPC1. It is important to note that in most cases, immunoprecipitation of STIM1 by either FLAG-antibody or STIM1 antibody is reduced following PQ treatment for a long time (24 hours), indicating that the C-terminal region of STIM1, which the antibody targets, is affected by PQ, probably due to a conformational change or the residency of the PQ molecule in STIM1. Taken together, PQ most likely targets STIM1 to mediate extracellular calcium influx.

**PQ association with STIM1 facilitates STIM1 binding to TRPC1 as well as the increase in intracellular calcium levels.** To examine that PQ associates with STIM1 for the STIM1-TRPC1 interaction as well as the increase in intracellular calcium levels, we then generated STIM1 point mutants as predicted in Fig. 3d. As indicated in the docking model, the potential amino acids in the

C-terminal region of STIM1 for PQ association are Pro584, Val585, Tyr586, V611 and Gly614. Pro584 and Tyr586 could associate with PQ via π-π interactions, Gly614 and Tyr586 could associate with PQ via electrostatic force, and V585 and V611 could associate with PQ via Van Der Waals Force. The insertion of PQ into the pocket formed by the above amino acids facilitates the continuous exposure of the poly-K region of STIM1 for association with D639&D640 residues in TRPC1 to gate extracellular calcium entry[21,33]. We therefore first examined whether mutagenesis of the D639&D640 residues in TRPC1 or the K778&K779 residues, the most critical amino acids in the poly-K region of STIM1 predicted to be modulated by PQ, would affect PQ-induced STIM1 association with TRPC1. As shown in Fig. 4a, b, consistent with our previous findings, PQ stimulation promotes STIM1 association with TRPC1. However, both TRPC1 D639A&D640A (DDAA) and STIM1 K778A&K779A (KKAA) fail to fully respond to PQ stimulation in terms of the STIM1-TRPC1 interaction (Fig. 4a, b). Moreover, FACS analysis revealed that ectopic expression of STIM1 or TRPC1 but not KKAA or DDAA enhances PQ-increased intracellular calcium levels (Fig. 4c, d). These results together suggest the importance of the D639&D640 residues in TRPC1 and the K778&K779 residues in STIM1 in mediating PQ toxicity.

Next, we generated three STIM1 mutants with the potential amino acids for PQ association mutated, including V585A&V611A (VVAA), P584A&Y586A (PYAA), and P584A&Y586A&G614A (PYGAAA). FACS analysis revealed that compared to wild-type

STIM1, PYAA and PYGAAA but not the VVAA mutant exhibit deficiency for PQ-induced intracellular calcium overload (Fig. 4e). To further confirm the observations of intracellular calcium overload, we performed single cell calcium imaging and found that compared to wild type STIM1 or the VVAA mutant, PQ-raised extracellular calcium entry is impaired in KKAA, PYAA, or PYGAAA expressing cells (Fig. 4f), indicating that P584 and Y586 are two essential residues in STIM1 to mediate PQ-induced STIM1 overactivation. Consistently, PQ-raised extracellular calcium entry is high in cells expression of STIM1 but not STIM1ΔK (Fig. 4f), further confirming the importance of the poly-K region of STIM1 in mediating PQ-raised extracellular calcium entry. Further analysis revealed that PQ-raised STIM1 association with TRPC1 is both blocked with mutagenesis of the P584&Y586 residues or the P584&Y586&G614 residues (Fig. 4g). These results together indicate that PQ mainly associates with P584&Y586 residues in STIM1, probably via π-π interaction. To address this possibility, we finally performed a pull-down assay by using PQ-Biotin in A549 cells ectopically expressing empty vector (EV), STIM1 or PYAA, and found that PQ-Biotin successfully associates with wild-type STIM1 but not the PYAA mutant. Furthermore, the association between PQ-Biotin and TRPC1 is also deficient in the PYAA expressing group (Fig. 4h), suggesting that STIM1 is the direct molecular target of PQ. Taken together, these results clearly show that PQ binds to STIM1 and thus promotes STIM1 association with TRPC1 to increase intracellular calcium levels.

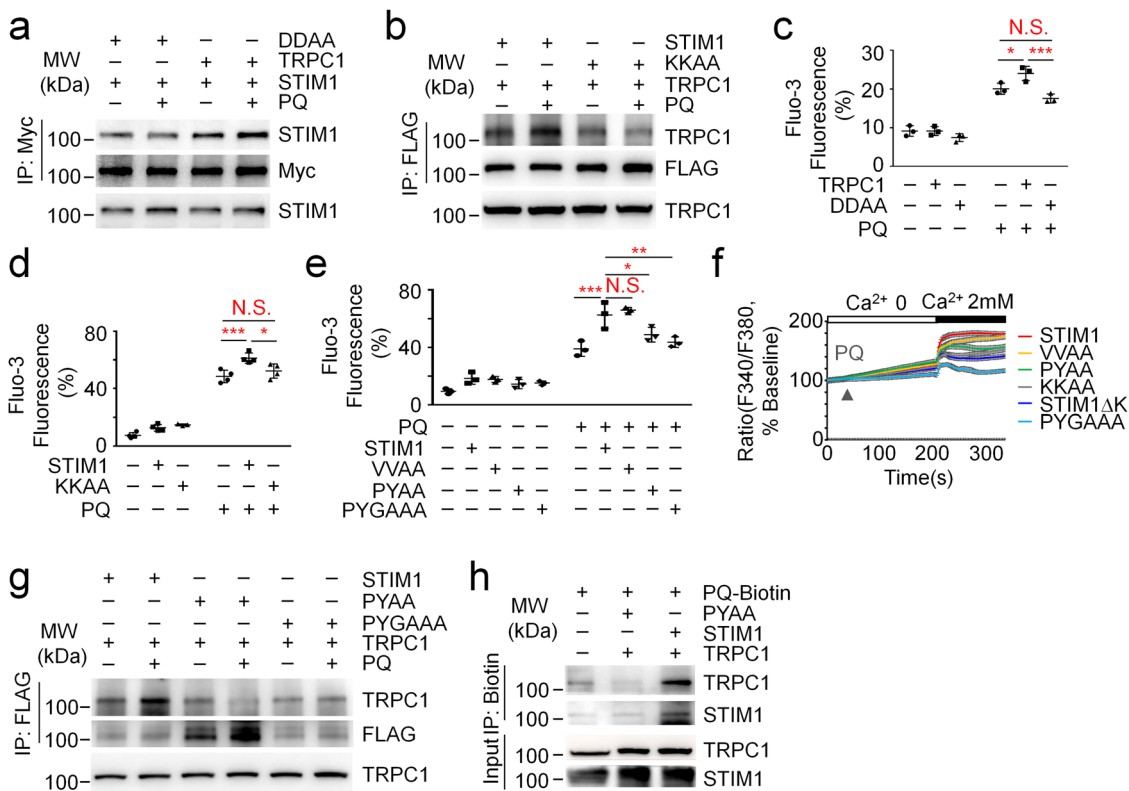

**Fig. 4 PQ association with the P584&Y586 residues of STIM1 promotes STIM1-TRPC1 interaction and increases intracellular calcium levels. a–b** Co-IP analysis of STIM1-FLAG association with TRPC1-myc or TRPC1 D639A&D640A-myc (DDAA) (**a**) or TRPC1-myc association with STIM1-FLAG or STIM1 K778A&K779A-FLAG (KKAA) (**b**) in A549 cells treated w/wo 800 µM PQ for 30 min. **c–e** FACS analysis of intracellular calcium levels by Fluo-3 staining in A549 cells with ectopic expression of TRPC1 or DDAA (**c**), STIM1 or KKAA (**d**), or STIM1, STIM1 V585A&V611A (VVAA), STIM1 P584A&Y586A (PYAA), or STIM1 P584A&Y586A&G614A (PYGAAA) (**e**). Cells were treated w/wo 800 µM PQ for 18 hours. Mean ± s.d., *P < 0.05; ***P < 0.001; N.S., not significant. Two-Way ANOVA. n = 3. **f** Calcium imaging of A549 cells infected with STIM1, STIM1ΔK, KKAA, VVAA, PYAA, or PYGAAA. Cells were treated with 800 µM PQ, followed by the addition of 2 mM CaCl₂. Mean ± sem. **g** Co-IP analysis of the TRPC1-myc association with STIM1-FLAG, PYAA-FLAG, or PYGAAA-FLAG in A549 cells treated w/wo 800 µM PQ for 30 min. **h** Pull-down analysis of PQ-Biotin association with STIM1 or PYAA in A549 cells. Cells were treated with 800 µM PQ-Biotin for 30 min. Immunoblot was probed using indicated antibodies.

**PQ promotes EMT and cell death in AT II cells mainly via the STIM1-TRPC1 axis**. Next, we wondered whether PQ-driven extracellular calcium influx is a critical mechanism for PQ toxicity. PQ has been well recognized to promote EMT in AT II cells and therefore lead to pulmonary fibrosis[6,34]. As extracellular calcium influx is essential for EMT in multiple cancer cells[11], we speculated that PQ-induced extracellular calcium influx would be a critical signal for EMT during pulmonary fibrosis. We then knocked down *Stim1*, *Orai1* and *Trpc1* in A549 cells (Fig. 5a-c) and analyzed the expression profiles of PQ-induced EMT markers. Consistent with previous reports[2], PQ significantly increases the expression of Vimentin, the mesenchymal marker, whereas decreases the expression of E-cadherin, the epithelial marker. Furthermore, such alteration in EMT markers is largely diminished in both shRNA-*STIM1* and shRNA-*TRPC1* infected A549 cells, but not in shRNA-*ORAI1* infected cells (Fig. 5a-f), indicating that the STIM1-TRPC1 axis but not the STIM1-ORAI1 axis is mainly responsible for PQ-induced EMT. It is worth noting that we repeatedly found a reduced expression of

E-cadherin in both STIM1- and TRPC1- deficient cells under physiological conditions, indicating that the STIM1-TRPC1 axis is also required for maintaining physiological functions in epithelial cells. Nevertheless, these results are consistent with our mechanistic insight that PQ is an agonist for STIM1 that facilitates STIM1 association with TRPC1. Except EMT, $Ca^{2+}$ overload is also well recognized to induce cell death[35], which is another well-established characteristic during PQ poisoning. Our results showed that *STIM1*, *ORAI1*, or *TRPC1* deficiency all significantly alleviates the cytotoxicity raised by PQ treatment (Fig. 5g), suggesting the importance of intracellular calcium burden for PQ-induced cell death.

## Discussion

PQ was once recognized as an excellent herbicide for agricultural production until it was found to be poisonous to mammals. PQ can be utilized safely by well following the manufacturer's guidelines. Most of the cases of PQ poisoning are due to self-administration or accidental ingestion, and unfortunately till now no efficient antidote

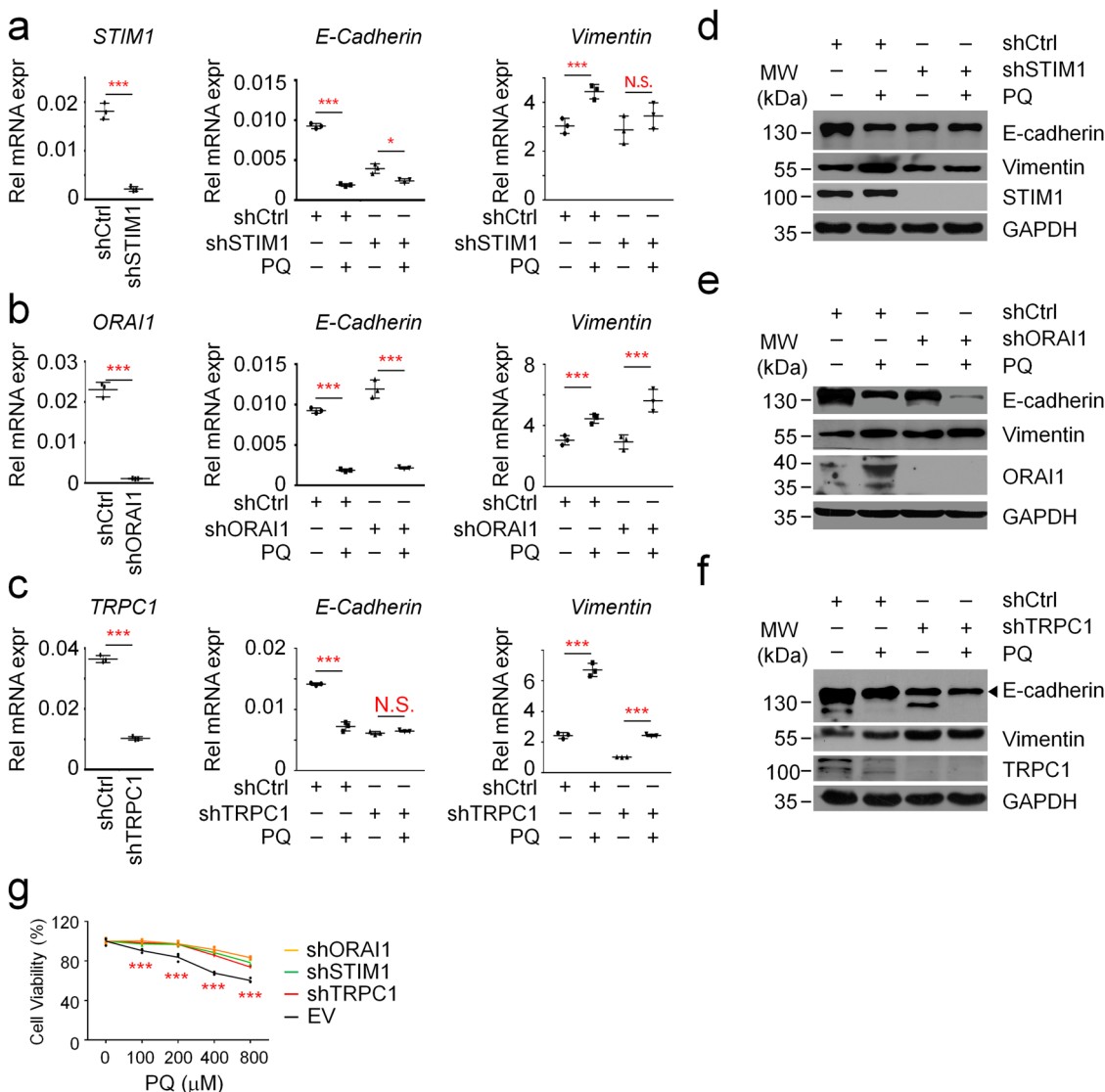

**Fig. 5 PQ induces EMT and cell death mainly via the STIM1-TRPC1 axis. a–c** RT-PCR analysis to detect the expression of *STIM1, ORAI1, TRPC1, E-Cadherin, or Vimentin* in A549 cells infected with shRNA-STIM1 (**a**), shRNA-ORAI1 (**b**), or shRNA-TRPC1 (**c**), respectively. The cells were treated w/wo 800 μM PQ for 24 hours. **d–f** WB analysis of the expression of E-Cadherin and Vimentin, makers of EMT, in A549 cells infected with shRNA-STIM1 (**d**), shRNA-ORAI1 (**e**), or shRNA-TRPC1 (**f**), respectively. The cells were treated w/wo 800 μM PQ for 24 hours. **g** SRB analysis of cell viability in A549 cells infected with shRNA-STIM1, shRNA-ORAI1, or shRNA-TRPC1. The cells were treated with 0, 100, 200, 400, or 800 μM PQ for 24 hours, as indicated.

has been developed to neutralize PQ toxicity[9]. For years, numerous efforts have been made to understand the mechanisms of PQ poisoning that reveals the importance of oxidative stress as one of the key mechanisms for PQ toxicity[8]; however, limited strategies have been identified to recover the patients. Here, we identified abnormal extracellular $Ca^{2+}$ entry as a critical upstream signal of PQ poisoning since the STIM1-TRPC1 association acts as the molecular target of PQ in pulmonary epithelial cells. STIM1-mediated extracellular calcium influx not only increases intracellular $Ca^{2+}$ levels but also facilitates oxidative stress[36,37]. In this scenario, PQ-increased intracellular $Ca^{2+}$ levels could be an important upstream signal of ROS, and thus strategies blocking PQ-induced extracellular $Ca^{2+}$ influx both prevent $Ca^{2+}$ increase and benefit the clearance of abnormal ROS levels in cells. In the clinic, charcoal perfusion and anti-inflammation strategies have been well established to treat PQ poisoning[38]. Therefore, further efforts are required to identify potential precise strategies targeting abnormal extracellular calcium influx or even combinational strategies with the established clinical treatments for PQ poisoning.

SOCE has been well recognized to be mainly activated by the association with STIM1 and ORAI1. Moreover, activation of TRPC1 by STIM1 also requires the participation of ORAI1 in the PM[32]. Though STIM1, ORAI1, and TRPC1 all mediate extracellular $Ca^{2+}$ entry, the cellular functions of STIM1-ORAI1 and STIM1-TRPC1 differ in multiple cell types[39]. In this study, we again found that although PQ-induced intracellular $Ca^{2+}$ overload in AT II cells could be diminished by STIM1, ORAI1, or TRPC1 deficiency, the PQ-induced EMT process is largely mediated by the STIM1-TRPC1 channel but not the STIM1-ORAI1 route. Consistent with our findings, a recent study of COVID-19 infection in AT II cells mediated by SOCE showed similar results, in which STIM1 deficiency is resistant to COVID-19 infection while ORAI1 deficiency show high susceptibility in AT II cells[40]. Though in this study the effect of TRPC1 has not been evaluated, it is possible that the STIM1-TRPC1 axis might be the major route for extracellular $Ca^{2+}$ entry that modulates COVID-19 infection in AT II cells. It is worth noting that we also found a significant reduction in blood $Ca^{2+}$ levels in COVID-19 infected patients, yet the levels are still in normal range. Nevertheless, these findings indicate the importance and difference between the STIM1-TRPC1 and STIM1-ORAI1 route in the modulation of cellular functions in AT II cells.

Our study further identified the importance of the poly-K region in STIM1 and the D639&D640 residues in TRPC1 in mediating PQ-raised STIM1 binding to TRPC1 as well as intracellular calcium increasement. The gating of TRPC1 by STIM1 has been revealed to be mediated by electrostatic interaction of D639&D640 residues in TRPC1 with the poly-K region of STIM1[21,23]. Intriguingly, in resting conditions, mutagenesis of the D639&D640 residues in TRPC1 or the K790&K791 residues in STIM1 (K684&K685 residues in STIM1 isoform 2) does not affect STIM1 association with TRPC1[21], whereas these mutants largely impair TG-raised STIM1 interaction with TRPC1[33], indicating that the association between the poly-K region of STIM1 and TRPC1 D639&D640 residues is promoted in pathological conditions. Here we do observe that PQ-raised STIM1 association with TRPC1 is impaired with mutagenesis of STIM1 K778&K779 residues or TRPC1 D639&D640 residues, unraveling two lysine residues in the poly-K region of STIM1 new for association with TRPC1 and gating extracellular calcium entry. Moreover, we further identified that PQ could associate with STIM1 via π-π interaction, which is predicted to facilitate the poly-K region in STIM1 association with D639&D640 residues in TRPC1 for intracellular calcium increase.

ROS production inside cells impacts numerous cellular functions[41]. Several upstream signals have been identified to generate ROS production whereas $Ca^{2+}$ is a special signal that not only promotes ROS production but could also be modulated by ROS. Under normal conditions, $Ca^{2+}$ promotes the Kreb cycle that consumes more oxygen and increases ROS levels, which is under fine-tune modulated[42]. However, when mitochondria are overloaded with $Ca^{2+}$, ROS production can be generated independent of the metabolic state[43]. Importantly, ROS levels are required for normal cellular functions and only high amount of ROS results in cell damage[41]. Therefore, PQ-induced STIM1-mediated calcium entry probably raises mitochondria $Ca^{2+}$ burden and might lead to abnormally high amounts of ROS accumulation, which could be another important source for PQ-raised ROS production, followed by EMT and cell damage. Moreover, although context dependent, ROS has been shown to stimulate extracellular $Ca^{2+}$ entry[37], ER $Ca^{2+}$ release via oxidation and activation of RyR[44], or even the SERCA pump via modulation of the redox state of its cysteine residues[45]. Therefore, it would be reasonable to expect that after long term exposure to PQ stimulation, the initial ROS production and $Ca^{2+}$ signal would promote each other and form a vicious cycle independent of PQ that results in severe cell damage, immune overactivation and lung injury, which could also explain why PQ-poisoned patients with depletion of blood PQ levels would still undergo disease progression. Also, PQ-raised ROS production elongates lifespan in C. elegans[46], whereas SOCE deficiency is not required for oscillatory $Ca^{2+}$ Signaling and ER $Ca^{2+}$ homeostasis in C. elegans[47], indicating that the vicious cycle between ROS and $Ca^{2+}$ signal does not exist in C. elegans and thus PQ shows beneficial effects. Nevertheless, blockage of the vicious cycle therefore is crucial for anti-PQ poisoning.

In conclusion, in this study we found that PQ targets the STIM1-TRPC1 axis for extracellular calcium influx following by intracellular $Ca^{2+}$ overload in pulmonary epithelial cells and thus results in pulmonary fibrosis. The identification of STIM1 as the molecular target of PQ not only indicates an unreported mechanistic insight into PQ toxicity but also reveals PQ as an agonist for STIM1 and thus activating extracellular calcium entry.

## Materials and methods

**Cell culture**. A549 cells, purchased from The Cell Bank of Type Culture Collection of Chinese Academy of Sciences, were cultured in Ham's F-12K (Kaighn's) Medium (Genom, GNM21127) supplemented with 10% fetal bovine serum (GIBCO,42Q1095K), penicillin (100 IU/ml, Genom, GNM15140) and streptomycin (100 μg/ml, Genom, GNM15140). MLE 12 cells were purchased from Anwei-Sci Biotechnology Co., LTD. (Shanghai, China) and cultured in Dulbecco's Modified Eagle Medium/Nutrient Mixture F-12 (DMEM/F12, AnWei-Sci) supplemented with 10% fetal bovine serum (GIBCO,42Q1095K), penicillin (100 IU/ml, Genom, GNM15140) and streptomycin (100 μg/ml, Genom, GNM15140). WI38 cells and HEK293T cells were cultured in Dulbecco's Modified Eagle Medium (DMEM, Shanghai BasalMedia Technologies Co., LTD., Shanghai, China) supplemented with 10% fetal bovine serum (GIBCO, 42Q1095K), penicillin (100 IU/ml) and streptomycin (100 μg/ml). Mycoplasma contamination was tested and all these cells were negative. These cell lines were grown at 37 °C in a 5% carbon dioxide incubator and were passaged following trypsinization.

**Quantitative RT-PCR**. A549 cells were infected with control-shRNA, STIM1-shRNA, ORAI1-shRNA, or TRPC1-shRNA and treated with 800 μM PQ for 24 h. The cells were lysed in TRIzol (Vazyme Biotech Co, R701-02) and total RNA was extracted and reverse transcribed into cDNA by using a HiScript III RT SuperMix kit (Vazyme Biotech Co, R323). The mRNA expression of STIM1, ORAI1, TRP family members, E-cadherin and Vimentin was determined by semi-quantification with ChamQ SYBR Color qPCR Master Mix (Vazyme Biotech Co, R323, Q411-02) in QuantStudio 7 Flex. Data were normalized to GAPDH and calculated by $2^{[-(Ct\ target\ gene-Ct\ GAPDH)]}$. The sequences of the PCR primers are provided in Supplementary Table 1 (Supplementary Table 1).

**Western Blot**. Cells were harvested and the proteins were extracted with ice-cold RIPA lysis buffer (Beyotime, P0013D). The protein concentration in the lysates was determined using a BCA protein assay kit (Thermo Fisher Scientific, 23225). Equivalent amounts of protein (25–40 μg) were separated by 10% SDS-PAGE gels and transferred onto a nitrocellulose filter membrane (NC) membrane (Pall, 66485) and blocked with 5% non-fat milk in Tris-buffered saline with Tween 20

(TBST). Primary antibodies were incubated at 4 °C overnight, including anti-FLAG (Abmart, M20008, 1:5000), anti-Myc (Affinity, T0052 1:5000), anti-STIM1 (Cell Signaling Technology, 5668, 1:3000), anti-ORAI1 (Santa Cruz, sc-377281, 1:500), anti-TRPC1 (Proteintech, 19482, 1:1000), anti-E-cadherin (Cell Signaling Technology, 3195 S, 1:1000), anti-Vimentin (Cell Signaling Technology, 5741 S, 1:1000), anti-GAPDH (Proteintech, 60004-1-1 g, 1:10000), anti-NFATc1 (Santa Cruz, sc7294, 1:500), and anti-NFATc2 (Santa Cruz, sc7296, 1:500).

**Immunofluorescence**. Cells were seeded on glass coverslips in a 24—well plate. The attached cells were then treated with 800 μM PQ or PQ-Biotin for 24 h as indicated, fixed in 4% paraformaldehyde for 10 min and permeabilized with 0.1% Triton X-100 (Biosharp, BS100) for 6 min at room temperature. The permeabilized cells were then blocked with 0.2% BSA for 30 min and incubated with primary antibodies for STIM1 (Cell Signaling Technology, 5668, 1:200) or NFATc1 (Santa Cruz, sc7294, 1:200) at RT for 1 h. Cells were then washed twice with PBS, followed by incubation with 488-conjugated goat anti-mouse secondary antibody (Thermo, A32723, 1:2000) or 488-conjugated goat anti-rabbit secondary antibody (Bioss, bs-0295G, 1:500) for 1 h. Fluorescent signals were imaged by laser scanning confocal microscopy (Leica).

**STIM1 Puncta analysis**. To analyze the spatial organization of STIM1, images were first collected based on immunofluorescence analysis of STIM1 in A549 cells treated w/wo PQ in a dose dependent manner, as indicated, for 24 hours. The images were then processed to select regions containing STIM1 positive signals by using Image-Pro Plus Image Analysis Software. The size and area of puncta were measured by pixel and calculated in μm or μm$^2$, respectively, as indicated (1 pixel = 1 μm).

**Reporter Gene assay**. To investigate the transcriptional activity of NFAT, we used a luciferase reporter gene assay. Briefly, A549 cells were transfected with NFAT-luciferase/Renilla plasmids by using Polyjet transfection reagent (Signagen, SL100688). The transfected cells were pretreated w/wo SKF in a dose dependent manner for 2 h, and then treated with 800 μM PQ for additional 24 h as indicated. The production of both NFAT-luc and Renilla were determined by using the dual-luciferase Kit Assay Reporter System (Vazyme Biotech Co, DL101-01) and examined by using a luminometer (Varioskan Flash, Thermo Fisher, MA, USA). The data are present as the ratio of the NFAT-luc luminescence to Renilla luminescence.

**Flow cytometric analysis**. A549 cells, MLE 12 cells or WI-38 cells were seeded in a six-well plate at a density of $1 \times 10^6$ cells/dish in complete growth medium. Following SKF treatment or not, the cells were exposed to PQ for 24 h, washed twice with PBS and incubated with 2 μM Fluo-3/AM (Beyotime, S1056) diluted in Hanks' Balanced Salt Solution (Beyotime, C0218) for 30 min at 37 °C in the dark. The stained cells were then dissociated with trypsin, resuspended with HBSS and collected to detect the presence of $[Ca^{2+}]_i$ using a FACScan flow cytometer (BD Accuri, NJ, USA).

**Single-cell Ca$^{2+}$ Measurements**. A549 cells were plated in 35 mm glass bottom dishes. After attachment, the cells were incubated with 2 μM Fura-2 AM (Beyotime, S1052) diluted in Ca$^{2+}$ free Hanks' Balanced Salt Solution (HBSS) for 30 min. The stained cells were treated with 1 μM TG or 800 μM PQ followed by the addition of 2 mM CaCl$_2$ for measurements of calcium fluxes, or as indicated. Fura-2 (340/380) filter set (pE-340fura, CoolLED, UK), a $20 \times 0.3$ N.A. objective lens, and a Photometrics Iris9 camera were used to capture images at a frequency of 1 image pair every 2 s. The relative fluorescence ratio at wavelengths of 340 nM and 380 nM (F340/F380) was measured by Visview and utilized for the assessment of cytoplasmic calcium levels.

**Immunoprecipitation**. To investigate the association between STIM1 and ORAI1 or TRPC1, A549 cells ectopically expressing STIM1 were treated with PQ for 24 h and lysed in RIPA buffer (Beyotime, P0013D) with the addition of protease inhibitor. The lysates were quantified by the bicinchoninic acid (BCA) method and incubated with FLAG-M2-beads (Abmart, M20018S, 10 μl/sample) or Myc-beads (Abmart, M20012, 10 μl/sample) as indicated for 3 h at 4 °C. The immune complexes were washed three times with PBS and subjected to Western Blot. For analyzing the association between TRPC1 and STIM1, the cell lysates were incubated with a TRPC1 antibody (1 μl/sample) overnight, followed by incubation with 10 μl protein A/G beads (Santa Cruz, SC-2003) for another 3 h at 4 °C. Specific antibodies were used for STIM1 (Cell Signaling Technology, 5668, 1:3000), ORAI1 (Santa Cruz, sc-377281, 1:500), or TRPC1 (Proteintech, 19482, 1:1000).

In order to examine the association between PQ and STIM1, TRPC1 or ORAI1, PQ was chemically modified with Biotin to generate PQ-Biotin (synthesized by Shanghai Uniray Biotech Inc.) and a pull-down assay utilized PQ-Biotin was performed. Briefly, PQ-Biotin was administrated to A549 cells similar as PQ for 24 h or the indicated time. The cells were then washed twice with PBS and lysed in RIPA buffer supplemented with protease inhibitor. The lysates were then incubated with streptavidin-agarose beads (Emd Millipore Co, 16–126, 15 μl/sample) for 3 h at 4 °C. The protein-containing beads were washed three times with PBS and subjected to Western Blot by using specific antibodies for STIM1 (Cell Signaling Technology, 5668, 1:3000), ORAI1 (Santa Cruz, sc-377281, 1:500), or TRPC1 (Proteintech, 19482, 1:1000).

**SRB assay**. A549 cells stably expressing empty vector, shRNA-STIM1, shRNA-ORAI1, or shRNA-TRPC1 were plated into 96-well plates at a concentration of $0.5 \times 10^4$ cells/dish in complete growth medium. The cells were treated with 0, 100, 200, 400, or 800 μM PQ for 24 h as indicated, fixed in 10% TCA for 60 min at 4 °C, rinsed with running water 5 times and stained with 0.4% SRB for 10 min at room temperature. The stained cells were then washed with 1% acetic acid 5 times to remove unbound SRB, blotted dry and solubilized in 50 μl 10 mM Tris base buffer. The solution was finally monitored on a microplate reader at 515 nm and the results are presented as the percentage of viable cells relative to SRB incorporation in control cultures (assumed to be 100% viable).

**Plasmids**. Human STIM1 cloned in PGMLV-CMV-MCS-3XFLAG-EF1-ZsGreen1-T2A-Puro was purchased from Genomeditech. Human ORAI1 plasmid was a kind gift from Dr. Youjun Wang. The cDNA clone encoding human TRPC1 was purchased from Sino Biological. All these three cDNA were subcloned into PLVX-IRES-Puro for further analysis. Overlap PCR was performed to generate STIM1 or TRPC1 point mutants. All constructs were confirmed by DNA sequencing.

**Lentivirus generation and cell infection**. HEK293T cells were plated in 6 cm dishes and transfected with 1 μg shRNA targeting STIM1 (targeting sequence CCTGGATGATGTAGATCATAA), ORAI1 (targeting sequence GAGTTACTCC GAGGTGATGAG) or TRPC1 (targeting sequence GCCCACCTGTAAGAAGA TAAT) using the puromycin-resistant pLKO.1 lentiviral vector, in the presence of 1 μg packaging plasmid (psPAX2) and 1 μg envelope plasmid (pMD2.G) using Polyjet transfection reagent. For overexpression of STIM1 and STIM1ΔK (residues 1-670 aa), STIM1 and STIM1ΔK were cloned into the puromycin-resistant pLVX-IRES-Puro lentiviral vector. HEK293T cells were transfected with 1 μg STIM1 or STIM1ΔK, together with 1 μg packaging plasmid (psPAX2) and 1 μg envelope plasmid (pMD2.G) using Polyjet transfection reagent. After 16 h, culture medium was refreshed and the cells were further incubated for 24 h. The supernatant containing lentivirus was then collected to infect A549 by mixture of 1 ml supernatant and 1 ml complete growth medium, together with 4 μg/ml polybrene for 24 h. Infected cells were selected by using 5 μg/ml puromycin for 36–48 h. Stable cell lines were generated by administration of 2.5 μg/ml puromycin for another two weeks.

**Statistics and Reproducibility**. Data are represented as the mean ± SD or mean ± SEM as indicated. The comparisons between any two groups were analyzed by two-tailed one type Student's *t*-test or in experiment with multiple comparisons were performed using one-way or two-way ANOVA, as indicated. The Pearson correlation coefficient was performed to analyze the correlation between electrolytes and PQ in blood. $*P < 0.05$, $**P < 0.01$, $***P < 0.001$. All of the experiments were performed with at least biological replicates or technical triplicates, as indicated.

**Reporting summary**. Further information on research design is available in the Nature Research Reporting Summary linked to this article.

## Data availability

All data of this study are available from the corresponding authors upon the reasonable request. Original full blots of western blots could be found in Supplementary Figure 2 (Figure S2a-S2o). Source data underlying the graphs is presented in Supplementary Data.

## Code availability

The structure of PQ utilized in this study is from 15938 (Pubchem). No available code is available for the C-terminal region of STIM1. The 3D structure of STIM1 modeling was performed using the software FR-t5-M[48] and I-TASSER[49]. STIM1 and PQ were docked into a complex by the software CB-Dock[50]. The model with the highest confidence among the top ten was selected by the docking score (Dock and Rerank[51]) and the expert's visual inspection.

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

## Acknowledgements

This work was supported by grants from the National Natural Science Foundation of China (81971555 to Z.Y., 81901951 to Y.Z., 81870053 to J.L., 82070058 to H.X., 82072210 to R.W.), Shanghai Pujiang Program (19PJ1408700 to Z.Y.), Shanghai Sailing Program (19YF1440100 to Y.Z.), Natural Science Foundation of Shanghai (20ZR1445200 to R.W.), and the National Key Research and Development Project (2020YFA0112900). We thank Shanghai Lu-Ming Biotech Co., Ltd. (Shanghai, China) for assistance with Metabolomics analysis.

## Author contributions

Z.Y. and W.Y. generated the concept, designed the experiments, analyzed the data and wrote the manuscript. W.Y. conducted the key experiments. Y.Z., X.Ma. And X.Meng performed the experiments. Z.Y. and R.W. interpreted the results and supervised the study. W.D. performed the docking and molecular modeling. Y.T., D.J., P.H., D.C., J.Z., R.T., J. L. and H.X. discussed and interpreted the results. All of the authors approved the final manuscript.

## Competing interests

The authors declare no competing interests.
