## [Peer Review File · Communications Biology]

Reviewers' comments:

Reviewer #1 (Remarks to the Author):

Paraquat (PQ) is a herbicide that is lethal to human. Better understanding of the cause that leads to pulmonary fibrosis is of great clinical relevance. In the manuscript entitled "Paraquat is an Agonist of STIM1 and Increases Intracellular Calcium Levels", Yang et al provided evidence indicating that Paraquat (PQ) works through STIM1-mediated calcium signaling. This is an interesting and important work. If the following points addressed, it would be more convincing.

- 1) Figure 2d, PQ-induced STIM1 puncta were not shown properly. After activation, STIM1 puncta are formed at cell periphery, or ER-PM junctions. Cellular images should be taken at the bottom of cells. If cell nucleus were clearly visible like those shown in Fig. 2d, the focus plane should be at the middle of cells. Under this situation, unless the cells are really flat, STIM1 fluorescence should show a bright, PM-like "circle" that outline the cell. Please show the images at the bottom of the cells as well.
- 2) Typical CRAC channels (STIM1-Orai1) are quite selective for calcium ions; while the so called SOC channels (STIM1-TRPC1) are less calcium selective. Since the authors indicate that STIM1-TRPC1 are mostly responsive for the PQ-induced calcium influxes. I would suggest to switch to Sr^{2+} at the end of Fig 2g. If it is typical CRAC channel, then the fura-2 response should diminish; if it is STIM1-TRPC, then the response should not change much. This assay would provide more supports for the author's claim.
- 3) The inhibitory effect of SKF96365 is quite weak. More pharmacological tests are needed. Either with higher SKF concentration, or use a different inhibitor. For example, add another fig that starts the same as Fig 2g, then add 50 μM 2-APB, and see whether the influx could be abolished. By showing a time course of SOCE inhibition, it will be more convincing that the PQ effect is a specific SOCE response.
- 4) Fig. 4 lacks a calcium imaging trace with TRPC1 with STIM1/STIM1-deltaK co-expression. SOCE is mediated by STIM1-Orai1, overexpression of Orai1 at high levels is known to inhibit SOCE by dominant negative effect. Compared with wild type cells, cells with co-expression of STIM1 together with TRPC1 should show dramatically increased PQ-induced Ca entry, while STIM1-deltaK+TRPC1 should not show such effect.
- 5) Along the same line as those in 4), calcium imaging traces with STIM1/TRPC1 knock down should be shown. If the authors claim were correct, then a dramatic reduction of PQ effect should be seen in KD cells. Results from 4) and 5) would provide very strong support for this STIM1-TRPC1 axis hypothesis.

Minor comments:

- 1) The PQ concentrations used in this study were very high, usually indicating non-specific effects. The authors should discuss why such high concentrations are reasonable, for example: PQ accumulation in the lung??
- 2) The changes in extracellular calcium levels of PQ patients are very interesting, but it is less likely caused by constitutive activation of STIM1-mediated Calcium influxes. As the increase in cytosolic calcium level is moderate, likely less than 1 micromolar (1/1000 mM), which cannot account for the sub mM level reduction of extracellular calcium level (100 folds larger). This effect on plasma calcium is likely due to problems in kidney. I would suggest take the last figure out, and leave it for further study.

Reviewer #2 (Remarks to the Author):

Yang W, et al. investigated the effect of paraquat (PQ) on STIM1 and STIM1-mediated SOCE. It may be useful for understanding the function and mechanism of PQ on STIM1 activation and EMT in AT II cells. Totally, the study is interestingly and the However, the manuscript is needed to careful revision

to promote the quality. I have below questions:

- (1) Many writing problems: for example the full names of ROS and NFAT are needed in the first time. WB in figure 5 legend. Hours should be shorted into hs.....
- (2) For clinical experiments, authors need to provide approve number of the ethics committee and general information of patients.
- (3) Please provide the data that have not been shown.
- (4) Many experiments are repeated in 3 times. The repeated number is too small and not enough. At least, the experiments are repeated lager than 5 times.
- (5) Why there are one or two bands of TRPC1 in different experiments?
- (6) In WB images, though authors labeled molecular weight, the position of the markers is needed to be labeled.
- (7) In Figure 5e and f, PQ treatment significantly suppressed TRPC1 expression but upregulated Orai1 expression. Therefore, authors need to provide the effect of PQ on cell proliferation, the expression of TRPC1, STIM1 and Orai1.
- (8) Authors claimed that STIM1 activity is related ROS production. However, authors did not provide any results of ROS production in overexpressing STIM1 mutant involving PQ effect.
- (9) The language polish and careful revision are needed.

Reviewer #3 (Remarks to the Author):

The manuscript proposes that paraquat (PQ) is a novel STIM1 agonist to activate TRPC1-mediated calcium entry, which in turn facilitates PQ-induced epithelial-mesenchymal transition (EMT). The authors further identified essential PQ-binding residues in STIM1 and demonstrated that the mutant failed to increase calcium entry by the treatment of PQ. Findings are of potential interest, however, the study suffers from unclear results obtained from biochemical analyses, especially western blot analyses, and inappropriate calcium measurements.

Major comments

- (1) Throughout the manuscript, qualities of data from western blot analyses including co-immunoprecipitation are poor. This makes results less convincing. In Figure 3a, STIM1 signals are unclear despite of immunoprecipitation using anti-STIM1 antibody. The same cases are acknowledged in Figure 2h for STIM, Figure 3b for TRPC1, and Figure 4b for STIM1-FLAG. In Figure 3c, there are no specific signals in STIM1(Δ K) and TRPC1. In addition, some panels showed a single band as TRPC1, whereas others showed double bands as TRPC1. I don't see any rules for these differences.
- (2) It is still unclear how PQ promotes TRPC1-mediated calcium entry. Figure 3F indicates that PQ preferentially binds to TRPC1 rather than STIM1, suggesting that PQ directly activates TRPC1 in a STIM1-independent manner. Where does PQ bind to TRPC1? Since it is unclear whether PQ-induced calcium entry depends on endogenous STIM1 and TRPC1 or not, it needs to be clarified by using Stim1- or TRPC1-deficient cells in Figure 5.
- (3) In this study, the authors measure intracellular calcium concentration by flow cytometer except a few panels. However, FACS analyses after 24 hr is not suitable for measurement of calcium entry by STIM1, as Fluo-3+ cell population may contain cells induced by both STIM1-dependent and - independent calcium entry. As did in Figure 2g, the effect of each mutant should be analyzed by a single cell-calcium imaging. If so, data will be more convincing.
- (4) The authors claimed that PQ promotes store-operated calcium entry (SOCE) based on Figure 2. However, this is not true. First, SKF-96365 only partially inhibited the number of Fluo-3+ cells. Second, the authors did not examine whether PQ enhances calcium entry after the treatment with thapsigargin (TG) in the presence of 2 mM calcium. Figure 2g do not indicate SOCE promotion by PQ since store-depletion is not induced.

(5) Since PQ treatment results in the accumulation of Fluo-3+ cells even after 24 hr stimulation, this suggests that PQ sustains rather than promotes calcium entry.

(6) It is unclear whether the relationship between reduced blood calcium level and PQ-promoted calcium entry in the body. The authors also stated that the reduction blood calcium level could be caused by various reasons. As Figure 6 is likely to make readers confused, Figure 6 should be omitted or used as a supplementary figure.

Minor comments

(1) Figure 5 does not contain any data related to cell death.

上海市第一人民醫院
SHANGHAI GENERAL HOSPITAL
上海交通大學附屬第一人民醫院
上海市紅十字醫院

上海交通大學醫學院
Shanghai Jiao Tong University School of Medicine

Zhengfeng Yang, Ph.D.,
Associate Professor,
Shanghai General Hospital,
Shanghai Jiaotong University, School of Medicine

Sep 1st, 2022

We would like to thank reviewers for their constructive comments and insightful suggestions on our manuscript. A point-by-point response to their comments is included below.

Reviewer #1:

Paraquat (PQ) is a herbicide that is lethal to human. Better understanding of the cause that leads to pulmonary fibrosis is of great clinical relevance. In the manuscript entitled “Paraquat is an Agonist of STIM1 and Increases Intracellular Calcium Levels”, Yang et al provided evidence indicating that Paraquat (PQ) works through STIM1-mediated calcium signaling. This is an interesting and important work. If the following points addressed, it would be more convincing.

1) Figure 2d, PQ-induced STIM1 puncta were not shown properly. After activation, STIM1 puncta are formed at cell periphery, or ER-PM junctions. Cellular images should be taken at the bottom of cells. If cell nucleus were clearly visible like those shown in Fig. 2d, the focus plane should be at the middle of cells. Under this situation, unless the cells are really flat, STIM1 fluorescence should show a bright, PM-like “circle” that outline the cell. Please show the images at the bottom of the cells as well.

Response: Thanks for the professional suggestion. As the reviewer pointed out, STIM1 puncta formation could be largely observed around endoplasmic reticulum - plasma membrane contact sites. We did capture the images close to the bottom of the cells, however, we could still observe cell nucleus. We therefore utilize thapsigargin (TG) as a positive control to analyze STIM1 puncta formation in A549 cells. As shown below, the bottom area of the cells is imaged and TG successfully induces STIM1 puncta formation, yet the nuclear signal could also be observed (Response Fig. 1A). As the reviewer commented, the morphology of the cells might be a potential reason to affect the observation of STIM1 puncta. We then imaged the above cells with 3D viewer and observed that the cells are flat and the nuclear is closely attached to the bottom area of the glass slides (Response Fig. 1B).

Based on the observations of TG-induced STIM1 puncta formation in A549 cells, we rechecked our preliminary data of Fig. 2d and confirmed that images mostly close to the bottom of the cells are selected for STIM1 puncta analysis. The confirmed images as well as the reanalyzed quantification results are now shown in the revised figure 2d & 2e. Also, the preliminary images are shown below (Response Fig. 1C).

Response Figure 1

2) Typical CRAC channels (STIM1-Orai1) are quite selective for calcium ions; while the so called SOC channels (STIM1-TRPC1) are less calcium selective. Since the authors indicate that STIM1-TRPC1 are mostly responsive for the PQ-induced calcium influxes. I would suggest to switch to Sr^{2+} at the end of Fig 2g. If it is typical CRAC channel, then the fura-2 response should diminish; if it is STIM1-TRPC, then the response should not change much. This assay would provide more supports for the author's claim.

Response: Thanks for the insightful suggestion. As the reviewer suggested, we performed single cell calcium imaging to measure if PQ stimulation would promote Sr^{2+} influx. As shown in Fig. S1d, PQ stimulation also activates Sr^{2+} influxes, indicating that PQ at least activates the STIM1-TRPC1 route.

3) The inhibitory effect of SKF96365 is quite weak. More pharmacological tests are needed. Either with higher SKF concentration, or use a different inhibitor. For example, add another fig that starts the same as Fig 2g, then add 50 μ M 2-APB, and see whether the influx could be abolished. By showing a time course of SOCE inhibition, it will be more convincing that the PQ effect is a specific SOCE response.

Response: Thanks for the professional suggestion. As suggested, we performed the calcium imaging with addition of 50 μ M SKF96365 after cells stimulated with 800 μ M PQ followed by 2 mM CaCl_2 . As expected, the PQ-raised calcium influx is largely reduced and normalized to the basal level, suggesting that PQ-induced calcium influx is mainly attributed to the STIM1-ORAI1 or the STIM1-TRPC1 route. We now added the result in Fig. 2k.

4) Fig. 4 lacks a calcium imaging trace with TRPC1 with STIM1/STIM1-deltaK co-expression. SOCE is mediated by STIM1-Orai1, overexpression of Orai1 at high levels is known to inhibit SOCE by dominant negative effect. Compared with wild type cells, cells with co-expression of STIM1 together with TRPC1 should show dramatically increased PQ-induced Ca entry, while STIM1-deltaK+TRPC1 should not show such effect.

Response: Thanks for the insightful suggestion. We now measured the calcium imaging in cells ectopic expression of STIM1 and TRPC1 or STIM1 Δ K and TRPC1, and found that PQ-raised extracellular calcium entry is high in cells expression of STIM1 and TRPC1 but not STIM1 Δ K and TRPC1, confirming the importance of the poly-K region of STIM1 in mediating of PQ-raised extracellular calcium entry. We further measured the STIM1 mutants in mediating PQ-induced extracellular calcium entry. Consistent with the Co-IP and FACS analysis shown in Fig.4, compared to wild-type STIM1 or the VVAA mutant, PQ-raised extracellular calcium entry is impaired in mutants exhibiting reduced interaction with TRPC1 stimulated by PQ, including KKAA, PYAA, and PYGAAA. These results together confirmed the importance of the poly-K region of STIM1 as well as the interaction between STIM1 and TRPC1 raised by PQ in modulation of extracellular calcium entry.

5) Along the same line as those in 4), calcium imaging traces with STIM1/TRPC1 knock down should be shown. If the authors claim were correct, then a dramatic reduction of PQ effect should be seen in KD cells. Results from 4) and 5) would provide very strong support for this STIM1-TRPC1 axis hypothesis.

Response: Thanks for the insightful suggestion. We now examined PQ-induced extracellular calcium entry in cells with STIM1, ORAI1, or TRPC1 deficiency. All the deficient cells exhibit largely diminished extracellular calcium entry compared to the empty vector expressing cells, suggesting the importance of STIM1, ORAI1 and TRPC1 in mediating the PQ-induced calcium influx. We and others show that ORAI1 is required for STIM1 and TRPC1 association, raising the possibility that PQ-raised calcium influx impaired in ORAI1 deficient cells could also be due to the deficiency of STIM1-TRPC1 association. Nevertheless, these results together indicate that PQ-raised extracellular calcium entry is mediated by the STIM1 and TRPC1 route. We now added the result in Fig. 2j.

Minor comments:

1) The PQ concentrations used in this study were very high, usually indicating non-specific effects.

The authors should discuss why such high concentrations are reasonable, for example: PQ accumulation in the lung?

Response: Thanks for the comment. It is true that compared to most of the compounds studied, the effective concentration of paraquat is very high. We and others (Kim et al., *Exp Mol Med*, 2011; Yamada et al., *PLoS One*, 2015; Zhu et al., *J Cell Mol Med*, 2016; Liu et al., *J Biochem Mol Toxicol*, 2022; Tian et al., *Sci Rep*, 2022) all reported that IC₅₀ of paraquat is around 800 μ M in A549 cells. The same concentration of paraquat also induces epithelial-mesenchymal transition in vitro, which is believed as one of the major reasons for paraquat-induced pulmonary fibrosis. Furthermore, we once reported that diquat, the analog of paraquat, could not induce TGF- β expression as paraquat does at the concentration of 800 μ M in A549 cells. Also, diquat with 800 μ M does not induce EMT as well (Yang et al., *Toxicol Res*, 2021). These observations together indicate that the high concentration of PQ induces a specific effect in A549 cells.

The potential reason for such high concentration of PQ to take effect might be relevant to the absorption and accumulation of PQ in pulmonary epithelial cells. PQ has been recognized to be accumulated mainly in the lung. PQ concentration rises progressively and highly in the lung while the plasma concentration of PQ is remained relatively constant during the first 30 h after administration of PQ. It has been believed that the early removal of PQ from plasma would be beneficial for reducing PQ accumulation into the lung and thus ameliorate toxicity, indicating a high accumulation of PQ into pulmonary epithelial cells is essential for PQ poisoning. Therefore, it would be reasonable to speculate that a high concentration of PQ accumulated into A549 cells is required for cell death and EMT process.

2) The changes in extracellular calcium levels of PQ patients are very interesting, but it is less likely caused by constitutive activation of STIM1-mediated Calcium influxes. As the increase in cytosolic calcium level is moderate, likely less than 1 micromolar (1/1000 mM), which cannot account for the sub mM level reduction of extracellular calcium level (100 folds larger). This effect on plasma calcium is likely due to problems in kidney. I would suggest take the last figure out, and leave it for further study.

Response: Thanks for the insightful comment. We agreed with the reviewer's comment that the reduced plasma calcium would be mainly ascribed to the problems in kidney. Whereas the constitutive activation of extracellular calcium entry into AT II cells would be an interesting possibility for the reduced plasma calcium levels that requires further efforts to elucidated, we will take the Fig.6 out from this study for further research in the future.

Reviewer #2:

Yang W, et al. investigated the effect of paraquat (PQ) on STIM1 and STIM1-mediated SOCE. It may be useful for understanding the function and mechanism of PQ on STIM1 activation and EMT in AT II cells. Totally, the study is interestingly and the However, the manuscript is needed to careful revision to promote the quality. I have below questions:

(1) Many writing problems: for example the full names of ROS and NFAT are needed in the first time. WB in figure 5 legend. Hours should be shorted into hs.....

Response: Thanks for the suggestions. We now modified the revised manuscript as the reviewer suggested. We further asked AJE to correct the writing problems. The editing certificate is shown below.

(2) For clinical experiments, authors need to provide approve number of the ethics committee and general information of patients.

Response: Thanks for the comment. The approved number of the ethics committee in this study is KYLL-2019-296. As suggested by the other two reviewers, we now remove the clinical data from the manuscript for further study.

(3) Please provide the data that have not been shown.

Response: The reduced plasma calcium in COVID-19 infected patients has also been reported in another study (Qian et al., QJM, 2020). The clinical data we observed are provided below.

(4) Many experiments are repeated in 3 times. The repeated number is too small and not enough. At least, the experiments are repeated lager than 5 times.

Response: Thanks for the comment. All the experiments are repeated at least three times. Some important results are further repeated more than 3 times as indicated, including STIM1 puncta formation analysis, FACS analysis and Co-IP analysis. We further repeated the Co-IP experiments to get more convincing results. Also, we performed single-cell calcium imaging to confirm the results of FACS analysis. As many other publications show the representative results with three

replicates (Zhu et al., Nat Commun, 2022; Horn et al., Nat Commun, 2022; Kumar et al., Nat Cell Biol, 2022; Wu et al, Nat Cell Biol, 2022; Lizumi et al., Commun Biol., 2022; Ueno et al., Commun Biol., 2022), we believed the results as well as the conclusions are convincing. We now added the new results in the revised manuscript.

(5) Why there are one or two bands of TRPC1 in different experiments?

Response: Thanks for the professional comment. We have utilized two TRPC1 antibodies in this study, including one from Affinity (Cat. No: DF12783) and the other one from Proteintech (Cat. No: 19482-1-AP). Both of two antibodies can detect two bands while the one from Proteintech can detect a major band. We confirmed the bands by examined TRPC1 in empty vector (EV) or shTRPC1 expressed cells (Response Fig. 2). Moreover, we repeated the WB results by using the TRPC1 antibody from Proteintech to reconfirm the results once generated by using the TRPC1 antibody from Affinity. For TRPC1 bands in Fig. 4, TRPC1 is ectopically expressed in A549 cells, making a clear one band of TRPC1. We sincerely apologize for the obscure results of TRPC1 and we now added the new results in our revised manuscript.

Response Figure 2

(6) In WB images, though authors labeled molecular weight, the position of the markers is needed to be labeled.

Response: Thanks for the comment. We now label the position of the markers in the revised manuscript.

(7) In Figure 5e and f, PQ treatment significantly suppressed TRPC1 expression but upregulated Orai1 expression. Therefore, authors need to provide the effect of PQ on cell proliferation, the expression of TRPC1, STIM1 and Orai1.

Response: Thanks for the suggestion. We now examined PQ-induced cell death in the condition with STIM1, ORAI1, or TRPC1 deficiency. As shown in Fig. 5g, STIM1, ORAI1, or TRPC1 deficiency all significantly ameliorates PQ-reduced cell viability, suggesting the importance of intracellular calcium burden for PQ-induced cell death. For the expression of TRPC1, STIM1 or ORAI1, we once screened the expression of calcium channels in the condition with PQ stimulation and found that ORAI1 is upregulated while TRPC1 is largely reduced with PQ treatment in A549 cells (Fig. 2a).

(8) Authors claimed that STIM1 activity is related ROS production. However, authors did not provide any results of ROS production in overexpressing STIM1 mutant involving PQ effect.

Response: Thanks for the suggestion. The correlation between STIM1 activation and ROS

production has already been reported in previous studies (Sokolowski et al., Redox Biol, 2020; Görlach et al., Redox Biol, 2015). As suggested, we also examined PQ-induced ROS production in conditions with or without SOCE deficiency or overexpressing STIM1 mutants. As shown in Response Fig. 3, consistent with previous reports, PQ significantly increases ROS production, while such increase is largely diminished in cells co-treated with SKF96365 (Response Fig. 3A), suggesting that PQ-raised extracellular calcium entry is essential for ROS production. Further analysis revealed that PQ-raised ROS production enhanced by ectopic expression of STIM1 is also normalized with mutagenesis of P584&Y586 residues (PYAA) or P584&Y586&G614 residues (PYGAAA) (Response Fig. 3B). Taken together, these results indicate that the enhanced STIM1 association with TRPC1 by PQ stimulation also promotes excessive ROS production.

Response Figure 3

(9) The language polish and careful revision are needed.

Response: Thanks for the suggestions. We now carefully reviewed the manuscript and asked AJE to further modify the manuscript. The editing certificate was shown above.

Reviewer #3:

The manuscript proposes that paraquat (PQ) is a novel STIM1 agonist to activate TRPC1-mediated calcium entry, which in turn facilitates PQ-induced epithelial-mesenchymal transition (EMT). The authors further identified essential PQ-binding residues in STIM1 and demonstrated that the mutant failed to increase calcium entry by the treatment of PQ. Findings are of potential interest, however, the study suffers from unclear results obtained from biochemical analyses, especially western blot analyses, and inappropriate calcium measurements.

Major comments

(1) Throughout the manuscript, qualities of data from western blot analyses including co-immunoprecipitation are poor. This makes results less convincing. In Figure 3a, STIM1 signals are unclear despite of immunoprecipitation using anti-STIM1 antibody. The same cases are

acknowledged in Figure 2h for STIM, Figure 3b for TRPC1, and Figure 4b for STIM1-FLAG. In Figure 3c, there are no specific signals in STIM1(Δ K) and TRPC1. In addition, some panels showed a single band as TRPC1, whereas others showed double bands as TRPC1. I don't see any rules for these differences.

Response: Thanks for the insightful comment. We apologize for the low quality of the Co-IP and WB results. We now modified the Co-IP protocol by increasing the amount of total protein lysates for Co-IP analysis (from around 250 μ g total protein to 500 μ g total protein, in another words, from 6cm dishes to 10cm dishes for protein lysates preparation for each sample). We now repeat all the WB results as pointed out.

For the issue of TRPC1 bands, we have utilized two TRPC1 antibodies in this study, including one from Affinity (Cat. No: DF12783) and the other one from Proteintech (Cat. No: 19482-1-AP). These two antibodies can both detect two bands, in which the one from Proteintech can detect a major band. We further confirmed the bands by examined TRPC1 in empty vector (EV) or shTRPC1 expressing cells (Response Fig. 2). We now repeated all WB analysis once utilized the TRPC1 antibody from Affinity by using the TRPC1 antibody from Proteintech. For TRPC1 bands in Fig. 4, TRPC1 is ectopically expressed in A549 cells, making a clear one band of TRPC1. We sincerely apologize for the obscure results of TRPC1 and we now added the new results in our revised manuscript.

Response Figure 2

(2) It is still unclear how PQ promotes TRPC1-mediated calcium entry. Figure 3F indicates that PQ preferentially binds to TRPC1 rather than STIM1, suggesting that PQ directly activates TRPC1 in a STIM1-independent manner. Where does PQ bind to TRPC1? Since it is unclear whether PQ-induced calcium entry depends on endogenous STIM1 and TRPC1 or not, it needs to be clarified by using Stim1- or TRPC1-deficient cells in Figure 5.

Response: Thanks for the insightful suggestion. We now examined PQ-induced extracellular calcium entry in cells with STIM1, ORAI1, or TRPC1 deficiency. All the deficient cells exhibit largely diminished extracellular calcium entry compared to the empty vector expressing cells, suggesting the importance of STIM1, ORAI1 and TRPC1 in mediating the PQ-induced calcium influx. We and others show that ORAI1 is required for STIM1 and TRPC1 association, raising the possibility that PQ-raised calcium influx impaired in ORAI1 deficient cells could also be due to the deficiency of STIM1-TRPC1 association. Nevertheless, these results together indicate that PQ-raised extracellular calcium entry is mediated by the STIM1 and TRPC1 route. We now added the result in Fig. 2j.

Considering that PQ-induced extracellular calcium entry is impaired in STIM1 mutants predicted and confirmed for PQ association, STIM1 at least is one of the targets of PQ. Whether PQ targeting

TRPC1 requires further effort to elucidate. So far, we fail to identify potential residues in TRPC1 highly possible for the direct association with PQ. Nevertheless, a crystal structure of PQ-driven STIM1-TRPC1 association would be a direct answer, which could be studied in the future.

(3) In this study, the authors measure intracellular calcium concentration by flow cytometer except a few panels. However, FACS analyses after 24 hr is not suitable for measurement of calcium entry by STIM1, as Fluo-3+ cell population may contain cells induced by both STIM1-dependent and -independent calcium entry. As did in Figure 2g, the effect of each mutant should be analyzed by a single cell-calcium imaging. If so, data will be more convincing.

Response: Thanks for the insightful suggestion. As suggested, we now performed single cell calcium imaging to measure the calcium imaging in cells ectopic expressing STIM1 mutants, including STIM1ΔK, KKAA, VVAA, PYAA, and PYGAAA. We found that PQ-raised extracellular calcium entry is high in cells expression of STIM1 but not STIM1ΔK, confirming the importance of the poly-K region of STIM1 in mediating PQ-raised extracellular calcium entry. Furthermore, consistent with the Co-IP and FACS analysis shown in Fig.4, compared to wild type STIM1 or the VVAA mutant, PQ-raised extracellular calcium entry is impaired in mutants exhibiting reduced interaction with TRPC1 stimulated by PQ, including KKAA, PYAA, and PYGAAA. These results together confirmed the importance of the poly-K region of STIM1 as well as the interaction between STIM1 and TRPC1 raised by PQ in modulation of extracellular calcium entry.

(4) The authors claimed that PQ promotes store-operated calcium entry (SOCE) based on Figure 2. However, this is not true. First, SKF-96365 only partially inhibited the number of Fluo-3+ cells. Second, the authors did not examine whether PQ enhances calcium entry after the treatment with thapsigargin (TG) in the presence of 2 mM calcium. Figure 2g do not indicate SOCE promotion by PQ since store-depletion is not induced.

Response: Thanks for the constructive suggestions. It seems that A549 is not very sensitive in response to the SOCE inhibitor as one study showed that low concentration of 2-APB (10 μM), another SOCE inhibitor, only transiently inhibits TG-induced SOCE in A549 cells while high concentration of 2-APB (75 μM) abolishes it (Padar et al., *Biochemical Pharmacology*, 2005). We therefore performed the calcium imaging with addition of a relatively high concentration of SKF96365 (50 μM) after cells stimulated with 800 μM PQ followed by 2 mM CaCl₂. The PQ-raised calcium influxes are largely reduced and normalized to the basal level with SKF treatment (Fig. 2k), suggesting that PQ-induced calcium influx is mainly attributed to the STIM1-ORAI1 or the STIM1-TRPC1 route.

Moreover, as suggested, we measured TG-raised extracellular calcium influx with or without PQ pre-stimulation, or PQ-induced calcium influx with or without TG pre-stimulation. Interestingly, we observed that PQ pretreatment enhances TG-induced SOCE process (Supplementary Fig. 1b) while TG pretreatment does not affect PQ-induced extracellular calcium influx (Supplementary Fig. 1c). We considered that TG pretreatment leads to a longer window for TG to take effect, which depletes ER calcium store and fully activates STIM1, making a later addition of PQ could not activate STIM1 anymore due to a saturated activation of STIM1 by TG. One evidence is that the intensity of SOCE activation stimulated by TG with PQ pretreatment (Supplementary Fig. 1b) is similar as the long term of TG stimulation (Supplementary Fig. 1c). These results further indicate that PQ also functions on STIM1 activation same as TG, yet with a different mechanism.

(5) Since PQ treatment results in the accumulation of Fluo-3+ cells even after 24 hr stimulation, this suggests that PQ sustains rather than promotes calcium entry.

Response: Thanks for the comment. We agree with the reviewer's point that PQ might also sustain the extracellular calcium entry. We have shown a transient stimulation of PQ could induce extracellular calcium influx (Fig. 2g, 2j, 2k, and 4h), suggesting that PQ promotes extracellular calcium entry. As PQ associates with STIM1 for STIM1-TRPC1 interaction, the long-term stimulation of PQ might therefore sustain the extracellular calcium influx that could not be auto-braked by store refilling as common SOCE do, which ultimately leads to PQ toxicity.

(6) It is unclear whether the relationship between reduced blood calcium level and PQ-promoted calcium entry in the body. The authors also stated that the reduction blood calcium level could be caused by various reasons. As Figure 6 is likely to make readers confused, Figure 6 should be omitted or used as a supplementary figure.

Response: Thanks for the insightful comment. We now take the Fig.6 out from this study for further research in the future.

Minor comments

(1) Figure 5 does not contain any data related to cell death.

Response: Thanks for pointing out the issue, we now added the results of cell viability in Fig. 5g. STIM1, ORAI1, or TRPC1 deficiency all significantly ameliorates PQ-reduced cell viability, suggesting the importance of intracellular calcium burden for PQ-induced cell death.

Reviewers' comments:

Reviewer #1 (Remarks to the Author):

As to my first comment about STIM1 puncta, I would say that if cells were treated and imaged carefully, one should be able to visualize the thin layer of ER that lies beneath nucleus. The logic is: From bottom up, there is PM, nuclear membrane and the nucleus. And the nuclear membrane is continuous with ER, where STIM1 is located. Thus there is definitely STIM1 beneath the nucleus. And in the response Fig. 1A, the authors did show some STIM1 distribution beneath the nucleus. Results in Fig. 1A clearly showed that this could be done by these authors.

The clearly visible nucleus (especially in PQ 400/800) was likely caused by sub-optimal focus at the bottom, thick optical slice (large pinhole) or some damages to the cells during immunostaining.

I would strongly suggest the authors replace their original images with better ones. Since this is not the major point of the manuscript, if the authors didn't have the best images of STIM1 puncta, it is also OK with me.

The authors have successfully addressed my other comments

Reviewer #2 (Remarks to the Author):

Authors answered my questions and I have no further questions.

Reviewer #3 (Remarks to the Author):

In the revised manuscript, the authors have properly addressed most of the issues I raised in the previous review. Additional new data make the manuscript much clearer and more comprehensive. However, I think that SOCE is not suitable for this manuscript because PQ does not elicit store depletion (Fig.2g). "STIM1-mediated calcium entry" is a more precise phrase. Overall, I am satisfied with the authors' sincere responses.

上海市第一人民醫院
SHANGHAI GENERAL HOSPITAL
上海交通大学附属第一人民医院
上海市红十字医院

上海交通大学医学院
Shanghai Jiao Tong University School of Medicine

Zhengfeng Yang, Ph.D.,
Associate Professor,
Shanghai General Hospital,
Shanghai Jiaotong University, School of Medicine

Sep 26th, 2022

We would like to thank reviewers for their careful revision and the insightful suggestions on our revised manuscript. A point-by-point response to their comments is included below.

Reviewer #1 (Remarks to the Author):

As to my first comment about STIM1 puncta, I would say that if cells were treated and imaged carefully, one should be able to visualize the thin layer of ER that lies beneath nucleus. The logic is: From bottom up, there is PM, nuclear membrane and the nucleus. And the nuclear membrane is continuous with ER, where STIM1 is located. Thus there is definitely STIM1 beneath the nucleus. And in the response Fig. 1A, the authors did show some STIM1 distribution beneath the nucleus. Results in Fig. 1A clearly showed that this could be done by these authors. The clearly visible nucleus (especially in PQ 400/800) was likely caused by sub-optimal focus at the bottom, thick optical slice (large pinhole) or some damages to the cells during immunostaining. I would strongly suggest the authors replace their original images with better ones. Since this is not the major point of the manuscript, if the authors didn't have the best images of STIM1 puncta, it is also OK with me.

Response: Many thanks for these professional comments on our STIM1 puncta data. We imaged cells with same procedures, and cells treated with PQ at 400 or 800 μ M do exhibit cellular damage (Kim et al., *Exp Mol Med*, 2011; Yamada et al., *PLoS One*, 2015; Zhu et al., *J Cell Mol Med*, 2016; Liu et al., *J Biochem Mol Toxicol*, 2022; Tian et al., *Sci Rep*, 2022). As suggested, we repeated the experiment and again found that PQ induces STIM1 puncta formation in a dose dependent manner. We now replaced STIM1 puncta data in the original Fig. 2d with the new data.

Reviewer #3 (Remarks to the Author):

In the revised manuscript, the authors have properly addressed most of the issues I raised in the previous review. Additional new data make the manuscript much clearer and more comprehensive. However, I think that SOCE is not suitable for this manuscript because PQ does not elicit store depletion (Fig.2g). "STIM1-mediated calcium entry" is a more precise phrase. Overall, I am satisfied with the authors' sincere responses.

Response: Thanks for the constructive comment. We agreed with the reviewer's point that PQ does not alter the ER store and promotes STIM1-mediated calcium entry. We do have this confusion

during our study and STIM1-mediated calcium entry is a prettily precise phrase for describing our results. We now modified our manuscript by using STIM1-mediated calcium entry.